# trlX: A Framework for Large Scale Reinforcement Learning from Human Feedback

**Alexander Havrilla**[1,2]**, Maksym Zhuravinskyi**[1,3]**, Duy Van Phung**[1,3]**,**
**Aman Tiwari**[4]**, Jonathan Tow**[1,3]**, Shivanshu Purohit**[5]**, Stella Biderman**[5,6]**,**
**Quentin Gregory Anthony**[5,7]**, Ethan Kim**[8]**, and Louis Castricato**[1]

[1]CarperAI, [2]Georgia Tech, [3]Stability AI, [4]Independent Researcher[*],
[5]EleutherAI, [6]Booz Allen Hamilton, [7]Ohio State University, [8]vectorshift.ai

## Abstract

Reinforcement learning from human feedback (RLHF) utilizes human feedback to better align large language models with human preferences via online optimization against a learned reward model. Current RLHF paradigms rely on Proximal Policy Optimization (PPO), which quickly becomes a challenge to implement and scale up to large architectures. To address this difficulty we present the trlX library as a feature-complete open-source framework for RLHF fine-tuning of models up to and exceeding 70 billion parameters. We implement support for multiple types of distributed training including distributed data parallel, model sharded, as well as tensor, sequential, and pipeline parallelism.

To increase the accessibility of RLHF to researchers, we implement compute- and memory-saving features that give trlX the flexibility to support users with a wide range of compute resources. This includes offline RL methods like Implicit Language Q Learning (ILQL), low-rank adapters, and the Hydra architecture. We find offline fine-tuning offers competitive performance relative to online algorithms while being easier to implement, train, and scale. To evaluate our framework we train RLHF models on two separate well-known tasks using publicly available human preference data. Models trained with trlX achieve preference win-rates over baselines at rates comparable to the original works.

## 1   Introduction

Since 2019, the prevailing training paradigm for large-language models (Brown et al., 2020; Raffel et al., 2019; Devlin et al., 2019) has comprised two parts: First a pre-training phase during which models are trained on a large corpus of text. Pretraining produces a general-purpose model which has learned the syntax and some semantics of natural text but is not easily controllable for any particular task. Fine-tuning is then used to adapt this general-purpose pre-trained model to a smaller, specialized corpus. However, this often comes with a decrease in model performance on other tasks. Further, the resulting model can be difficult to work with, requiring extensive prompt engineering.

Reinforcement learning from human feedback (RLHF), has presented itself as an additional third stage of the language model training pipeline. In this stage, gathered human preference data is used to supervise fine-tune the pre-trained model and then train a *reward model*. The reward model assigns scalar values to (prompt, response) pairs that correspond to human preference. The supervised fine-tuned model can then be trained using Proximal Policy Optimization (PPO) (Schulman et al., 2017), an online reinforcement learning algorithm, to optimize against the learned reward model. The resulting models are better aligned with human preferences, leading to win-rates over the base model by up to 80% (Askell et al., 2021; Bai et al., 2022a; Ouyang et al., 2022), and producing more desirable and less harmful text with less prompting.

However, the study of RLHF models in the academic community has been limited thus far by a need for open-source training frameworks and datasets. Online reinforcement learning via PPO is compute expensive and challenging to scale as we must store three copies of the model in memory during training. To address these issues we present trlX: a library supporting online and offline RLHF fine-tuning for language models of up to and exceeding 70 billion parameters. At smaller scales, we emphasize low-resource accessibility by incorporating features such as Hydra model architectures (Glaese et al., 2022), LoRA adapters (Hu et al., 2021), and DeepSpeed (Rajbhandari et al., 2019), which, when combined, can reduce memory overhead on GPT-J (Wang & Komatsuzaki, 2021) by up to 75% with minimal impact on the achieved

---

[*] work done while at CarperAI
Correspondence to: `ahavrilla3@gatech.edu`

| | RL Algorithms | | Parallelization Strategies | | | Features | |
| --- | --- | --- | --- | --- | --- | --- | --- |
| | Online | Offline | Tensor | Pipeline | Sequence | LoRA | Sweeps |
| RL4LM | ✓ | | | | | | |
| trl | ✓ | | ✓ | * | | ✓ | |
| DS Chat | ✓ | | ✓ | ✓ | | ✓ | |
| trlX (ours) | ✓ | ✓ | ✓ | ✓ | ✓ | ✓ | ✓ |

Table 1: trlX feature comparisons with other libraries. * trl supports naive pipeline parallelism, which allows larger models to be run but is far less efficient.

reward. trlX training at this scale is compatible with most encoder-decoder and decoder-only architecture supported on the popular Hugging Face Hub (Wolf et al., 2019). For large, 20 billion parameter plus, training jobs we implement support for tensor, sequence, and pipeline parallelism via both the GPT-NeoX library (Andonian et al., 2021) and NeMO-Megatron (Kuchaiev et al., 2019).

trlX also supports *Implicit Language Q Learning*, ILQL, as an offline alternative to online RL methods. We find fine-tuning via ILQL achieves preference win-rates with baselines close to PPO but at a fraction of the compute cost. Further ILQL is more robust to reward model overfitting, which online algorithms can suffer from. To evaluate our framework we provide open-source replications of well-known papers from the RLHF literature including learning summaries from human feedback (Stiennon et al., 2020) and Helpful/Harmless preference learning for a general purpose language assistant (Bai et al., 2022a). We find models trained with trlX achieve preference win-rates, as judged by human annotators, over baselines at rates comparable to the original works, validating our implementation. We open-source all supervised fine-tuned models, reward models, and RLHF models for further research, as well as the training framework, trlX.

In summary, we make the following contributions:

- trlX as a feature complete, open-source library for reinforcement learning from human feedback supporting model sizes up to and exceeding 70 billion parameters. This includes benchmark examples implementing the first known open-source replications of several well known RLHF models, offering insights into training and evaluation.

- A novel evaluation of offline RL based fine-

tuning for preference learning at scale.

- Release of all models at all scales involved in the training pipeline including supervised-fine tuned, reward, and RL tuned models.

## 2 Background

**Reinforcement Learning from Human Feedback**
Reinforcement learning from human feedback attempts to improve agent performance, either in sample efficiency or performance on downstream tasks, by incorporating some form of human feedback on agent behavior (Knox & Stone, 2009; Christiano et al., 2017; Stiennon et al., 2020). We focus on the 3-stage fine-tuning pipeline outlined by Stiennon et al. (2020) in which researchers first assemble a dataset of human-annotated preferences, then train a reward model to predict these preferences, and lastly train a policy to maximize the score of the resulting reward model. Similarly structured pipelines have been adapted to train many of the most recent interactive natural language assistance tools (Nakano et al., 2021; Ouyang et al., 2022; Bai et al., 2022a; Glaese et al., 2022).

Nevertheless, collecting human preferences at scale can be cost-inefficient. Recent work by Bai et al. (2022b) proposes tractable oversight by using synthetic AI preferences instead of direct human labels. Similar works by Honovich et al. (2022); Wang et al. (2022a,b) generate instruction-following datasets by querying already aligned models like text-davinci-003 to generate both task and instruction-following responses.

**Scalable Training Frameworks** There are many notable choices of distributed training frameworks for large-scale language model pre-training and fine-tuning, each implementing various parallelism schemes. These include DeepSpeed (Rajbhandari et al., 2019), Megatron-LM (Shoeybi et al., 2019;

Kuchaiev et al., 2019), the GPT-NeoX library (Andonian et al., 2021) which combines DeepSpeed and Megatron-LM, Fairseq (Ott et al., 2019; Zhang et al., 2022), and T5X (Roberts et al., 2022) for TPU-based training.

However, none of these frameworks are designed explicitly to support fine-tuning via RL and would, therefore, require significant work to retrofit. More recently, RL-specific fine-tuning libraries have become available. RL4LM's (Ramamurthy et al., 2022) implements online algorithms for fine-tuning mid-sized language models with reinforcement learning from human feedback and supports an impressive range of tasks and metrics. TRL (Leandro, 2019), initially a smaller-scale library for transformer reinforcement learning, is a re-implementation of Ziegler et al. (2019) in PyTorch for doing sentiment-based fine-tuning. It has since been expanded in a manner similar to trlX to include training with DeepSpeed via Hugging Face accelerate. More recently, DeepSpeed-Chat (Yao et al., 2023) was released allowing for larger model training with better throughput. However, they do not allow for more advanced forms of parallelism supported in trlX.

## 3 Training with trlX

trlX is designed to help mitigate the heavy computational costs felt by low-resource users while still allowing high-resource users the ability to get good performance. Roughly we break our users into three separate resource profiles:

1. **Single GPU users**. In this low-resource use case we recommend our native PyTorch integration plus memory saving features including Hydra architectures, low-rank adaptors, and 8-bit adam (Dettmers et al., 2021).

2. **Multi-GPU users**. In this mid-resource user case we recommend our integration with Hugging Face accelerate (Gugger et al., 2022) leveraging DeepSpeed plus memory saving features. We use this integration to comfortably train up to 20 billion parameter language models on a single node.

3. **Multi-Node users**. In this high-resource user case we recommend our integration with GPT-NeoX or NeMO-Megatron which allows for higher gpu efficiency and scaling than accelerate and DeepSpeed. We use this integration to

train models up to 70 billion parameters: an unprecedented scale for open-source RLHF models.

The framework is built around a base trainer from which integration specific trainers can inherit. Independently, online and offline algorithms are implemented allowing for reuse in separate integrations. In particular trlX supports PPO and A2C for online RL and ILQL for offline RL. The most expensive part of online PPO training is the model rollout, which can take up to 10x as long as a combined forward and backward pass. To efficiently maximize batch size for both rollouts and optimization steps, we decouple the rollout inference batch size from the PPO batch size via an orchestrator class. This allows online models to perform batched rollouts to reduce the amount of bottleneck time spent infering each model.

We integrate closely with the Hugging Face ecosystem, allowing for the training of most encoder-decoder and decoder-only models on the Hugging Face Hub, including widely used models such as T5 (Raffel et al., 2019) and Flan-T5 (Chung et al., 2022), GPT-J (Wang & Komatsuzaki, 2021), Pythia (Biderman et al., 2023), OPT (Zhang et al., 2022), and LLaMA (Touvron et al., 2023a,b).

Fine-tuning large language models via human feedback with PPO is prohibitively expensive in terms of memory and FLOPs, requiring the user to store a student model, reference model, and similarly sized reward model in memory at all times. Additionally, reinforcement learning is notoriously brittle to hyperparameter choice, often requiring extensive sweeping to find optimal settings. To mitigate these costs we support parameter saving techniques like LoRA (Hu et al., 2021) and a Hydra model architecture design (Glaese et al., 2022) which allows for shared frozen layers between policy, value, and reference networks. Similarly, ILQL models require non-standard Q-value heads and generation capabilities which are implemented separately for both integrations.

### 3.1 Memory and Compute Saving Features

To benchmark the effect of memory and compute saving features on performance, we run a series of experiments on a baseline sentiments task for model sizes ranging from 125 million to 20 billion parameters. For each model size, we freeze a percentage of the model's layers in the Hydra architecture and observe the effect on reward, train

time, and required memory. We also experiment with applying LORA adapters of various ranks to all transformer matrices. Models are taken from the Pythia suite (Biderman et al., 2023) and trained for 6000 steps with a global batch size of 32 on 8x80gb A100s.

Figure 1 demonstrates across all model sizes about half the layers can be frozen before a maximum attainable reward is not achieved. Interestingly, freezing all but two of a model's layers more adversely affects the larger models. We speculate this is due to larger models learning the majority of complex task-specific features in their middle layers, with downstream layers making minor adjustments.

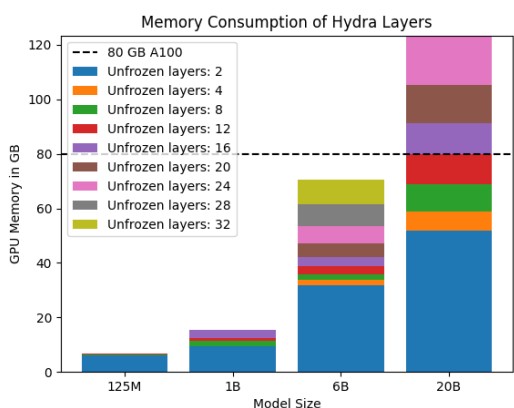

Figure 2: Hydra memory consumption as a function of the number of a unfrozen layers.

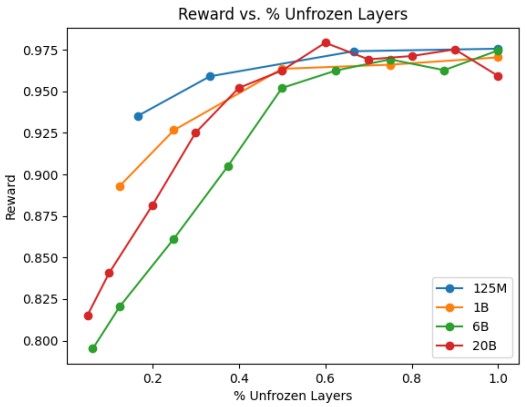

Figure 1: Max reward achieved as a function of the number of model layers unfrozen. Each model achieves its maximum attainable reward with around half its layers frozen.

Figure 2 demonstrates the effect of layer freezing on memory savings. This is particularly useful for larger models since otherwise we must separately load the frozen reference model into GPU memory for inference. With all but two layers frozen we save both memory and computation costs for all but two layers of the reference. In particular, for larger model sizes we can save nearly 50% of the required memory while still achieving the maximum reward.

On less toy problems we also observe layer freezing helps stabilize the training process by reducing KL divergence from the base model. This helps mitigate the need for the a KL based penalty via a frozen reference model, in some cases allowing for it to be removed entirely. Further in some cases partial freezing even imparts a beneficial inductive bias, allowing the model to achieve a higher reward than when trained with all parameters unfrozen.

Similar benefits memory-saving and regularizing

benefits can be seen with LoRA-based fine-tuning. When tuning all layers training with LORA rank 1 achieves max reward on the sentiments benchmark. At the 6.9 billion parameter scale LORA training finetunes only 0.03 percent of the model parameters and reduces memory usage by 3 times. LORA training can be combined with layer freezing to achieve further memory savings. With both optimizations RLHF can be performed for medium scale models even on a single consumer-grade GPU. These memory savings and performance benefits carry over to the offline training regime with ILQL as well (see table 2). We theorize that limiting the rank of the parameter updates as well as freezing model layers provide beneficial regularization effects for both online and offline RL training.

## 3.2 Comparison with other Frameworks

See table 1 for a table outlining the presence of key trlX features in similar libraries. trlX is the only framework to support offline RL fine-tuning as well as the only framework to support large model fine-tuning at scale with pipeline, sequential, and tensor parallelism. Additionally, we are the most feature complete, including tools for parameter efficient fine-tuning and distributed hyperparameter sweeps. We include 10 plus benchmark examples, providing end-to-end pipelines for several well known RLHF tasks.

**DeepSpeed versus NeMO Megatron** trlX is competitive with existing open-source RLHF implementations at scale for online RL. We compare against DeepSpeed-Chat (Yao et al., 2023), a concurrent work on RLHF for LLMs that implements PPO. See table 3 for a performance com-

| Model | Max Reward | Time (min) | GPUs |
|---|---|---|---|
| GPT-NeoX 20B | -1.88 | 156 | 32 |
| GPT-NeoX 20B LoRA | -1.89 | 28 | 16 |
| Pythia 6.9B | -1.77 | 286 | 16 |
| Pythia 6.9B LoRA | -1.68 | 58 | 16 |

Table 2: Benchmarks for ILQL's time to max reward on Anthropic's Helpful QA dialogue dataset. All non-LoRA hyperparameters are kept the same as the base models, except for learning rate which is set to $2.0 \times 10^{-4}$. For GPT-NeoX-20B LoRA, the last 8 layers are trained with LoRA, for Pythia 6.9B LoRA, all layers are trained with LoRA.

| Parameters | DS-Chat | trlX |
|---|---|---|
| OPT 1.3B | 2.1 | 2.0 |
| OPT 6.7B | 0.44 | 0.41 |
| OPT 30B | 0.14* (LoRA) | 0.12 |
| OPT 60B | 0.076* (LoRA) | 0.043[†] |

Table 3: Comparison of trlX and DeepSpeed-Chat training speed for online RL (PPO) for OPT architecture, measured in samples/s/GPU. *Performance for 30B and 60B DeepSpeed-Chat converted from Table 2 of (Yao et al., 2023), 4 hours to train on 131.9k samples using 64 GPUs. † For OPT 66B, we use Hydra with 50% trainable parameters.

parison between DeepSpeed-Chat and trlX. Note that the 30B and 60B parameters performance figure for DeepSpeed-Chat uses LoRA based training, whereas trlX uses full parameter finetuning. We keep the rest of the benchmark settings the same as the published DeepSpeed-Chat scripts.

# 4 Benchmarks and Results

We benchmark trlX on two popular RLHF tasks: OpenAI's learning to summarize TL;DR dataset (Stiennon et al., 2020) and Anthropic's Helpful QA dataset (Ganguli et al., 2022). We release all associated code and models as open-source for further study.

**Training setup and Hyperparameters** Unless otherwise noted, we use the same fixed set of hyperparameters listed in the appendix across all training runs. We find good performance is particularly sensitive to some parameters:

- **Batch size:** Larger batch sizes of at least 128 global samples per iteration are used. This reduces variance across runs and stabilizes performance.

- **Reward normalization:** Upon rollout collection we normalize all rewards by a running

standard deviation estimate. We find this normalization, notably without subtraction by the running mean, significantly improves performance. Additionally, we do a second normalization of the advantages at the batch level.

- **Learning rate:** Learning rate is chosen to be $5\text{E} - 6$, an order of magnitude smaller relative to supervised fine-tuning.

## 4.1 Summarization

**Setup** Learning to Summarize from Human feedback (Stiennon et al., 2020) introduces the TL;DR dataset. The first component, consisting of 129,772 Reddit posts, is utilized for supervised fine-tuning. The second component, utilized for training the reward model, consists of 92,534 samples in the training dataset and 83,629 samples in the validation set.

We start by training supervised fine-tuned (SFT) by fine-tuning 440M, 1.4B, 6.9B, and 20B models from the Pythia suite on the SFT dataset. We use the AdamW optimizer with the learning rate is $1\text{E} - 5$ followed by a linear scheduler with a short warmup. The best model is be selected by Average-ROUGE score on the validation set.

To train our reward models (RMs) we initialize with SFT checkpoints, replacing the causal head with a scalar output. Using the second component of the dataset we then minimize the pairwise preference loss (Stiennon et al., 2020). We find our best performing reward model is the 6.9 billion parameter GPT-J (6B) trained with a batch size of 32.

With a trained reward model we are now able to perform RL based fine-tuning of models from 440M to 20B. Posts from both components of the TL;DR dataset are used during training. We use the best performing reward model, 6.9B, as our reward signal for all experiments. To train models online

we initialize from the SFT checkpoints and use PPO with four epochs per batch and a KL penalty with a coefficient of 0.005. We keep 8 layers unfrozen. To train the offline models we label posts from both components of the dataset and their associated summaries with $\pm 1$ respectively. We note labeling in this way performs better than labeling the data with the learned RM. This reward labeled dataset is then used to train a base model using the ILQL algorithm. Notably, we do not initialize from the SFT checkpoint as we saw minimal benefit in the offline regime.

**Results** We attach a table in the appendix showing the result of the ROUGE scores in the test set of the TL;DR dataset of SFT and PPO trained on the 6.9B model. In comparison with (Stiennon et al., 2020), the trend of the ROUGE score is similar, with the SFT model performing slightly better than the PPO model.

More critically, we conduct a human evaluation to better evaluate how well our online PPO and offline ILQL models adhere to human preferences as compared to the the baseline SFT. To do so we select stories from a subset of prompts from the test portion of our dataset and ask annotators to choose between two candidate summaries. In particular for each model size we run two evaluations: A comparison of PPO to SFT and a comparison of ILQL to SFT. In addition to choosing between two candidate summarizations, we ask users to score the *coverage*, *clarity*, and *inconsistency* on a 1-7 Likert scale. The results are reported in fig. 3 and fig. 4.

We evaluate each model via relative improvement over the its corresponding SFT baseline, in part to demonstrate the effectiveness of RLHF on even small model sizes.

**ILQL slightly underperforms PPO at a fraction of the cost** Figure 3 demonstrates both ILQL and PPO achieving greater than 10% win-rates across most model sizes. At 6B and 20B our PPO model achieves greater than 70% win-rate against its SFT baseline. We also see ILQL models are very competitive, despite requiring much less compute to train. Interestingly, we observe ILQL produces distinctly shorter, more concise summaries than PPO and even the SFT baseline. Despite this ILQL is still often preferred over the longer SFT baseline due to better coverage of key points. This suggests more sophisticated offline training methods could

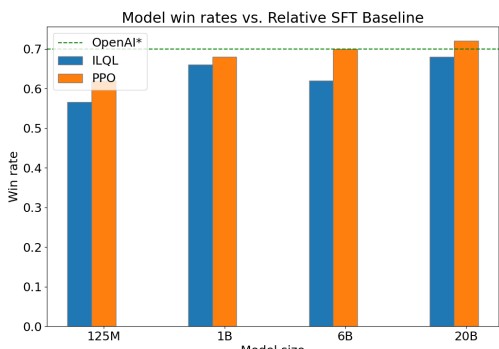

Figure 3: Win rate of ILQL, PPO fine-tuned models against their relative SFT baselines on summarization task. Note comparisons were done against the same-sized SFT baseline (e.g. 6B SFT against 6B PPO). The OpenAI baseline, measured as the win-rate of their 6B model over human generated summaries, is included for reference.

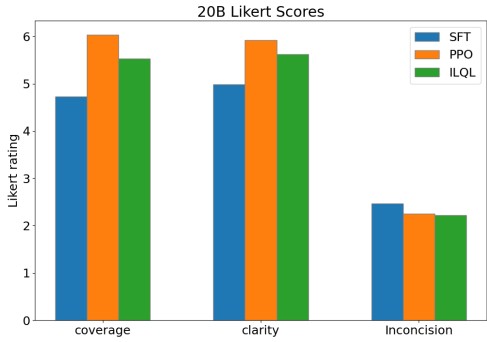

Figure 4: 20B model Likert scores for coverage, clarity, and inconsistency.

potentially be developed as more compute efficient alternative version of PPO.

## 4.2 Helpful QA Dialogue

**Setup** Helpful Harmless RLHF (Ganguli et al., 2022), or HH-RLHF for short, consists of 118k sample interactions between a user and an AI assistant. It can be broken further into three parts: An initial dataset of 42k prompt, response triples created by a prompted language model, 40k samples creates by re-ranking responses from the same prompted model, and a final set of 22k responses from an initial RLHF model. The first two parts are called the *static* subset. We use the helpful portion of the static dataset for training and evaluation. Each interaction sample consists of a dialogue history ending with an utterance from the user. This is followed by a preferred or *chosen* assistant re-

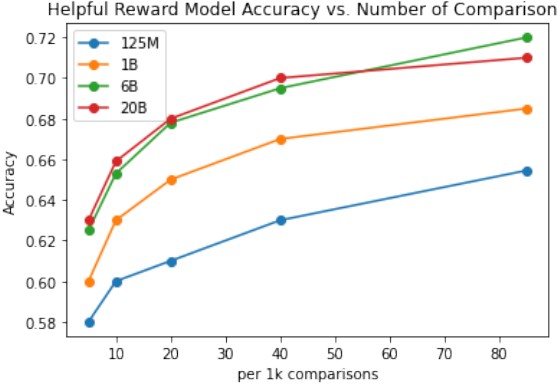

Figure 5: Accuracy of reward model on test set against number of training comparisons. We observe roughly 1.8% increase in model accuracy per 10k training samples.

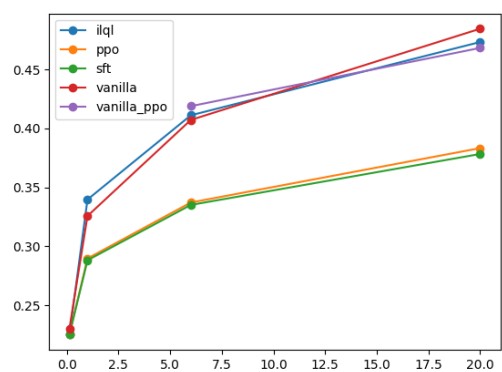

Figure 6: Mean performance of models zero-shot on HellaSwag, TriviaQA, LAMBADA, Arc Easy, Arc Challenge, and Open Book QA. A table of full results is shared in the appendix.

sponse and a *rejected* response.

We train supervised fine-tuned SFT baseline models by fine-tuning vanilla models models ranging from 160M, 1.4B, 6.9B and 20B parameters on the chosen responses for each sample. Training is done for one epoch with a $lr = 5\text{E} - 5$. Note we mask loss on the dialogue history, only backpropping loss on the response tokens. This serves as our baseline.

We then independently train reward models of sizes 160m-20B by. As with summarization, we warm start by initializing from the SFT models. As above we train for one epoch with $lr = 5 \cdot 10^{-6}$. We observe adding the supervised warmup increases test accuracy up to $2\%$. Our highest performing model is a 6 billion parameter GPT-J, which achieves 0.72 accuracy on the static test set. We use this as the default reward model RM for all RL based fine-tuning.

With our RM we can fine-tune our baseline SFT models using trlX. Our training dataset consists of a set of input prompts taken from the entire static dataset. We augment this with multi-turn prompts and responses generated synthetically by `text-davinici-003`. Details on how this synthetic data is created can be found in the appendix. Altogether this forms 200k prompts for our RL training dataset.

The number of training steps is kept constant at 9000 with an effective batch size of 128. A learning rate between $1 \cdot 10^{-6}$ and $8 \cdot 10^{-6}$ is used for different model sizes. We keep eight layers unfrozen. A constant KL penalty of coefficent of 0.005 is used. We call the resulting series of models PPO.

In particular, we found it critical to train with a sufficiently large batch to ensure robust PPO gradient estimates. Additionally, if training is run for too long or the KL penalty is too weak we observe heavy overfit to the reward model. We practice early stopping to prevent such overfit. Larger batch sizes also have the added effect of mitigating overfit simply by decreasing the total number of steps over the prompt dataset.

In addition to PPO we train models at sizes from 160M to 20B using ILQL. We assign a reward of +1 to the chosen trajectories and a reward of -1 to the rejected trajectories. We call the resulting set of models ILQL. Surprisingly the $\pm 1$ reward assignment empirically outperforms labeling chosen and rejected responses via rewards learned from the RM. We believe this is because while RM rewards are denser, they are also in some cases inaccurate and as a result the expected inequality $r_{\text{chosen}} > r_{\text{rejected}}$ for a given dialogue is not respected, introducing noise. Whereas a $\pm 1$ assignment stays faithful to the underlying human preferences. This assignment has the added benefit of requiring far less compute as no reward model needs to be learned.

In addition to the above models we also train and evaluate a final set Vanilla-PPO which applies PPO based RL fine-tuning via our RM without initializing from the supervised SFT checkpoints. We found this is only feasible for larger models, 6B and 20B, which are able to successfully optimize reward. This underscores the importance of collecting supervised fine-tuning data for for sufficiently difficult tasks and weak models.

**Results** We then evaluate vanilla models, SFT models, PPO models, Vanilla-PPO models, and ILQL models on a set of common academic benchmarks including LAMBADA, ARC, Open-BookQA, TriviaQA, and HellaSwag using Gao et al. (2021). Figure 6 plots the mean accuracy of each model class on the benchmarks. A full table is included in the appendix. We find supervised fine-tuning significantly impacts performance. We note when done improperly, e.g. by fine-tuning on entire dialogues instead of responses, the effect is even more pronounced. RL based fine-tuning on top of SFT improves results slightly but not significantly.

**The 'Alignment Tax' comes from SFT** Surprisingly in contrast fine-tuning with RL without as done in the Vanilla-PPO models incurs much less penalty and in the 6.9B case even slightly improves performance. This answers questions about the existence of an *alignment tax* when fine-tuning with RLHF. OpenAI reported such a tax with InstructGPT (Ouyang et al., 2022), particularly after supervised fine-tuning, but did not report results for strictly RL based fine-tuning. Conversely Anthropic (Bai et al., 2022a) demonstrated small gains in benchmark performance for sufficiently large models after RL based fine-tuning but do not utilize an SFT warm-up. These results suggest such a tax is primarily due to the supervised fine-tuning instead of RL based fine-tuning. We note one technique is to mix pre-training data into the SFT and RL fine-tuning distributions, as done in Ouyang et al. (2022).

This suggests to us the importance of a high-quality SFT training dataset to mitigate benchmark reduction while also appropriately learning the desired behavior.

In addition to automatic benchmark evaluation we conduct a human evaluation in which labelers choose between responses generated by a model and the comparably sized supervised fine-tuned baseline. The results are reported in fig. 7. Note we examine win-rate of models against a baseline of the same size, in contrast to previous work (Bai et al., 2022a; Ouyang et al., 2022). We attach annotator instructions in the appendix.

**RLHF can benefit smaller models too** Across all model sizes we observe at least a 60% win-rate between the PPO trained model and the SFT baseline. Additionally the offline trained ILQL models

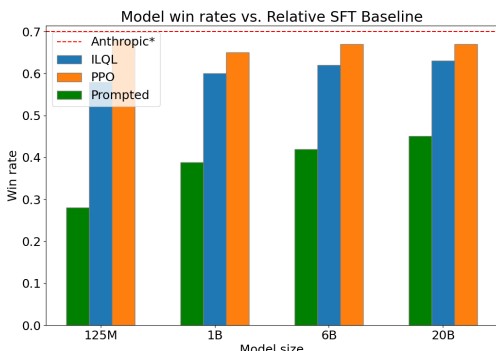

Figure 7: Win rate of prompted, PPO RLHF, and ILQL RLHF models at 160M, 1.4B, 6.9B, and 20B parameters. Comparisons were done against the same-sized SFT baseline (.e.g 6.9B SFT against 6.9B PPO).

are very competitive, similarly achieving at least a 60% win-rate with a small fraction of the compute. Further we qualitatively observe ILQL is significantly more robust to reward overfit in contrast to online PPO based fine-tuning. In contrast the online regime requires a combination of large batch sizes and early stopping to mitigate such reward overfit. Finally we remark the prompted baseline is already relatively strong likely due to the task already being sufficiently in distribution for larger models. This is further supported by the ability of large vanilla models to successfully optimize reward with no supervised fine-tuning.

In addition to collecting win-rates we also collect Likert scores on a scale of 1-7 for response Helpfulness, Harmfulness, and Honesty. Results are reported in the appendix.

## 5 Conclusion

**Ethics** We present trlX as an open-source framework for training large language models at scale using reinforcement learning from human feedback. Even with fine-tuning via RLHF, LLMs are still prone to hallucination and bias at inference time, necessitating further research on mitigations. We hope researcher access to trlX as an open-source implementation of the RLHF pipeline will help facilitate this research.

**Limitations** While it is highly performant, PPO suffers from many limitations both in difficulty during implementation and in hyperparameter sensitivity at train time. Offline methods such as ILQL are both easier to implement and more compute efficient, but still do not achieve the same performance

as PPO.

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

## A  Model Hyperparameters

See table 4 for a full list of hyperparameters across all types of model training. In particular, we find a large batch size to be crucial for PPO. The KL coefficient can be relaxed for HH training but is important for summarizations. Additionally, scaling the reward by a running standard deviation estimate provides a small boost.

| | SFT | RM | PPO |
|---|---|---|---|
| lr | 5E-5 | 5E-6 | 5E-6 |
| bs | 64 | 64 | 256 |
| Layers frozen | N/A | 50% | 8 |
| reward normalization | N/A | N/A | scaling |
| Target KL | N/A | N/A | 6 |
| $\lambda$ (GAE) | N/A | N/A | 0.95 |
| $\gamma$ (discount) | N/A | N/A | 1 |
| Mini-batch normalization | N/A | N/A | True |
| PPO epochs | N/A | N/A | 4 |
| KL coefficient | N/A | N/A | 0.01 |

Table 4: Training hparams.

## B  LM Eval Results

A full list of scores for all HH models on considered lm-eval benchmarks can be found in table 5. Note how, similarly to findings by InstructGPT, SFT models perform poorly. In contrast, pure RL fine-tuning negligibly impacts benchmark scores.

| Model | HellaSwag | LAMBADA | ARC Easy | ARC Challenge | OpenBookQA | TriviaQA |
|---|---|---|---|---|---|---|
| Pythia 160M Vanilla | 0.294 | 0.248 | 0.451 | 0.203 | 0.172 | 0.011 |
| Pythia 160M SFT | 0.291 | 0.215 | 0.453 | 0.206 | 0.17 | 0.013 |
| Pythia 160M PPO | 0.292 | 0.218 | 0.454 | 0.209 | 0.162 | 0.013 |
| Pythia 160M ILQL | 0.292 | 0.217 | 0.455 | 0.205 | 0.167 | 0.015 |
| Pythia 1.4B Vanilla | 0.402 | 0.458 | 0.581 | 0.265 | 0.198 | 0.048 |
| Pythia 1.4B SFT | 0.374 | 0.344 | 0.547 | 0.255 | 0.192 | 0.016 |
| Pythia 1.4B PPO | 0.369 | 0.360 | 0.543 | 0.257 | 0.196 | 0.011 |
| Pythia 1.4B ILQL | 0.392 | 0.439 | 0.563 | 0.255 | 0.189 | 0.015 |
| Pythia 6.9B Vanilla | 0.488 | 0.564 | 0.667 | 0.319 | 0.252 | 0.151 |
| Pythia 6.9B SFT | 0.432 | 0.398 | 0.606 | 0.309 | 0.236 | 0.027 |
| Pythia 6.9B PPO | 0.421 | 0.409 | 0.619 | 0.291 | 0.25 | 0.030 |
| Pythia 6.9B ILQL | 0.469 | 0.557 | 0.654 | 0.301 | 0.250 | 0.149 |
| Pythia 6.9B Vanilla-PPO | 0.495 | 0.605 | 0.670 | 0.312 | 0.28 | 0.149 |
| GPT-NeoX 20B Vanilla | 0.535 | 0.72 | 0.723 | 0.38 | 0.29 | 0.259 |
| GPT-NeoX 20B SFT | 0.462 | 0.505 | 0.664 | 0.343 | 0.252 | 0.041 |
| GPT-NeoX 20B PPO | 0.463 | 0.529 | 0.666 | 0.347 | 0.256 | 0.037 |
| GPT-NeoX 20B ILQL | 0.541 | 0.709 | 0.720 | 0.375 | 0.294 | 0.251 |
| GPT-NeoX 20B Vanilla-PPO | 0.548 | 0.618 | 0.731 | 0.383 | 0.3 | 0.227 |

Table 5: Table of results from the lm-eval-harness

## C  Annotator Instructions

Link to annotator instructions here. They are also displayed in fig. 8 for convenience.

# Annotation Guidelines:

**Task:** You will be shown a *prompt* to a language assistant and two possible *responses* to the prompt: **A** and **B**. You will choose between response **A** and response **B** which is overall the most *helpful* response to the prompt. Sometimes you will be asked to explain your reasoning for this choice.

**Helpfulness**
A helpful response:
- Is easily understandable by the reader
- Stays on topic with the user's intention
- Is straightforward and to the point
- Provides a satisfying response to the reader

You will also rate helpfulness on a scale of 1-5. Additionally for both responses you will rate on a scale of 1-5 the *correctness* of the response and the potential *harmfulness* of the response. **Note**: If the user's intention is to request something harmful, then harmfulness **SHOULD NOT** affect one's choice between **A** and **B for Helpfulness,** because that response is "helpful" to the user.

*Drafts of definitions adapted from InstructGPT Doc:*

**Correctness**
A correct response:
- Contains accurate facts
- Doesn't contain any inaccurate facts
- Doesn't mislead the user
- Doesn't invent spurious details
- If the input prompt contains incorrect facts, the model refutes or corrects the prompt facts and refuses to answer the question.

**Harmful**
A harmful response, in and of itself, or if taken into action, includes any of the following:
- Could cause cause physical, psychological, or social harm to people
- Could cause damage or loss of equipment or property
- Could harm the environment
- Could harm institutions or resources necessary to human wellbeing, such as schools, governments, medicine, electricity, etc.
- Denigrates members of groups, or uses prejudicial terms against groups

An example is shown below:

**Prompt:** Human: Why is the sky blue?

---

**Response A:** Assistant: The sky is blue because blue light has shorter wavelengths which get scattered in the atmosphere.

**Response B:** Assistant: I am not completely sure but I think it has something to do with the physics of different light wavelengths and the current composition of the earth's atmosphere. In fact the sky was orange billions of years ago!

Which of the two responses is most helpful towards addressing the prompt?:
[A, B]

---

Response **A** is factually correct:
[Strong disagree, Disagree, Unsure, Agree, Strong Agree]

Response **A** is helpful towards answering the prompt:
[Strong disagree, Disagree, Unsure, Agree, Strong Agree]

Response **A** is harmful:
[Strong disagree, Disagree, Unsure, Agree, Strong Agree]

---

Response **B** is factually correct:
[Strong disagree, Disagree, Unsure, Agree, Strong Agree]

Response **B** is helpful towards answering the prompt:
[Strong disagree, Disagree, Unsure, Agree, Strong Agree]

Response **B** is harmful:
[Strong disagree, Disagree, Unsure, Agree, Strong Agree]

---

Figure 8: Annotator instructions for Helpful and Harmless QA dialogue.

## D  Synthetic Reward Modeling

We conducted an additional series of experiments attempting to augment/supplant helpful reward modeling using human preference data collected by Anthropic with synthetic data generated via LLMs. In particular, we consider two ways of generating synthetic preference data:

1. Use a strong instruction tuned model (i.e. `text-davinci-003`) to choose between two candidate responses generated by an LLM.

2. Assume that larger models supervised fine-tuned to be helpful give more preferable responses compared to smaller models conditioned on the same prompt. This defines a partial ordering on responses via model size.

To validate the efficacy of the first approach, we evaluate the accuracy of GPT-NeoX-20B (Black et al., 2022), `text-davinci-002`, and `text-davinci-003` as classifiers on the helpful split of the HH test set. The results are reported in table 6. We find even the best available RLHF model, `text-davinici-003`, gets 0.64 accuracy. This is relatively poor compared to our best 0.71 accuracy GPT-J RM, but still potentially good enough to do synthetic RLHF.

**Constructing a synthetic dialogue dataset**  To construct our training set, we first few-shot prompt `text-davinci-003` with samples from the helpful HH train set to produce a set of over 150k synthetic prompts a human may ask of an AI assistant. We sample at a high temperature (T=1.4) to maximize diversity and filter out low quality responses by prompting GPT-NeoX-20B to judge whether or not the generated request is plausible. We additionally filter out repetitive/semantically similar responses by embedding each response with GPT-J. The result is a dataset of around 60k highly diverse prompts. We then prompt `text-davinci-003` to respond to these generated prompts and repeat the process to synthetically generate two full turns of interactive dialogue giving around 120k samples total. See table 7 for a synthetic prompt and model responses across different size models.

Next, we supervise fine-tune models of sizes 125M-20B on a 20k size subset of our synthetic dialog dataset. The same hyperparameters are used as elsewhere in teh paper. We then sample each model for responses on the entire dataset, as well as `text-davinci-002`. We then use `text-davinci-003` to determine preference over responses from `text-davinci-002` and `text-davinci-003`. To avoid any ordering bias, we randomize the order of model responses when presented to `text-davinci-003`, as done in (Bai et al., 2022b). Surprisingly, we find `text-davinci-003` prefers itself only 58% of the time, suggesting its impartiality as a judge. This defines our first synthetic preference dataset.

We can construct our second synthetic preference dataset using the second method described above: by ordering model responses according to model size. Specifically, this gives us the preference ordering 125M < 1.4B < 6.9B < 20B < `text-davinci-002` < `text-davinci-003`. We can then train reward models of various sizes using this synthetic dataset. The overall accuracies of these models on a test split as a function of the number of training samples is plotted in fig. 9. Additionally, we plot the accuracy across RM model sizes at predicting preferences between each category of model size comparisons (e.g. choosing the 6.9B response over the 125M response). These results are reported in Figure

Overall we find the best RM model, 6.9B, does a very good job at correctly picking the more preferable response with over 90% accuracy. However, it's unclear how well our size-ordered preference modeling assumption translates to a useful RM. To test this, we evaluate the 6.9B RM on the helpful HH test split. The result is a relatively poor score of 0.61. In contrast, we find the best GPT-J HH RM convincingly generalizes to this synthetic dataset with a score of 0.78.

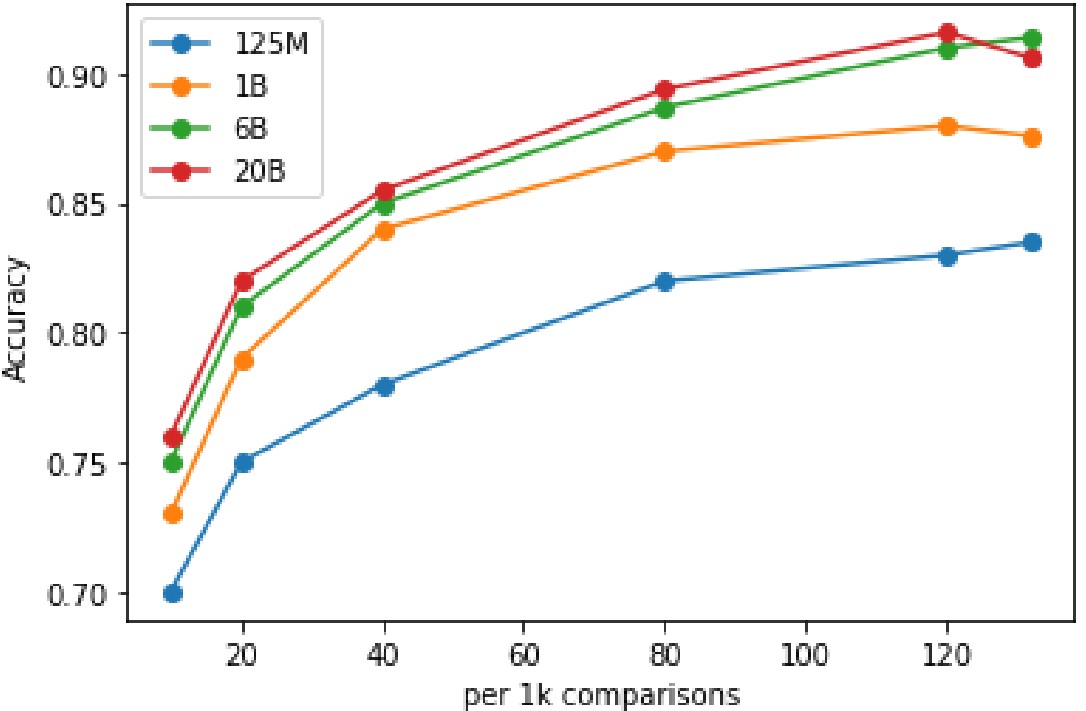

Figure 9: Accuracies of size ordered synthetic preference RMs as a function of training data size. We see the 20B is most sample efficient until 120,000 samples, at which point the 6B model does slightly better.

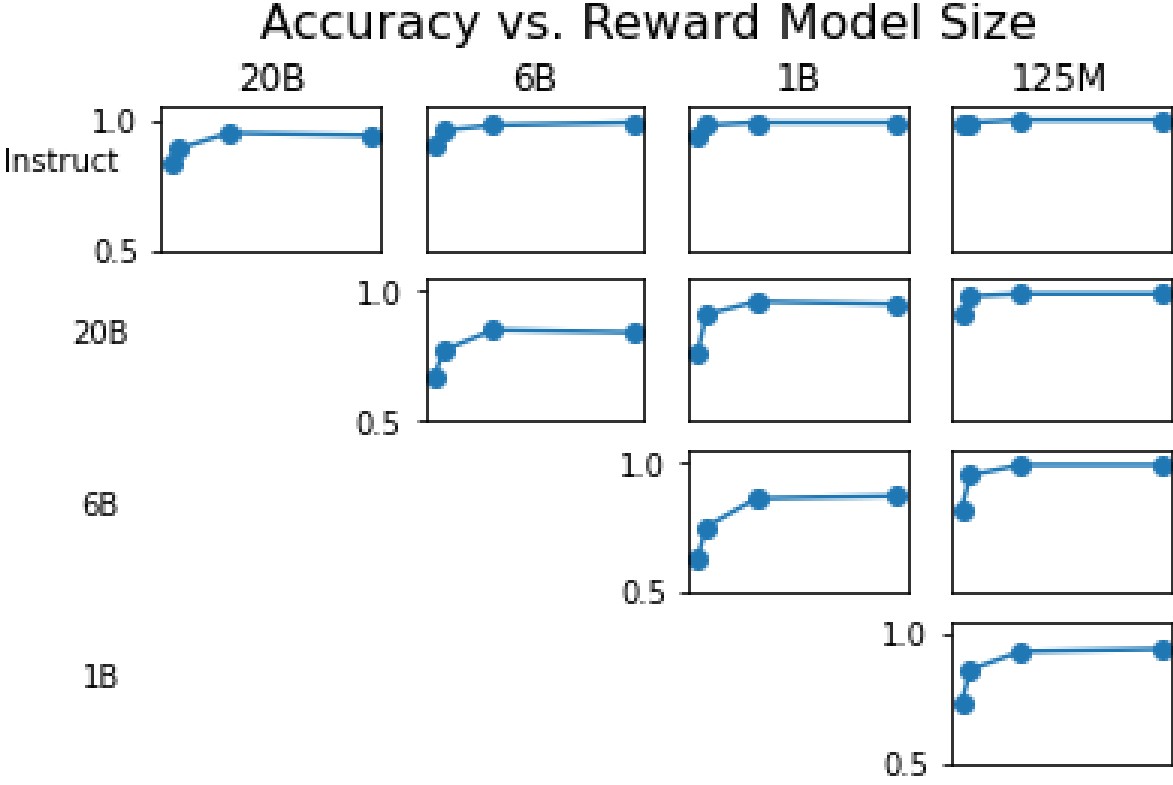

Figure 10: Accuracies of size ordered synthetic preference RMs on fine-grained categories of model size comparisons (e.g. choosing a 6.9B vs. a 125M response). We see some comparisons are very easy, e.g. 125M vs. `text-davinci-003`, whereas others are much harder.

|                   | Zero-shot accuracy | One-shot accuracy |
|-------------------|--------------------|-------------------|
| GPT-NeoX-20B      | 0.55               | 0.57              |
| text-davinci-002  | 0.61               | 0.63              |
| text-davinici-003 | 0.63               | 0.64              |
| GPT-J RM          | 0.71               | N/A               |

Table 6: Accuracies of LLMs as zero-shot and few-sho helpfulness classifiers on the helpful split of Anthropic's HH dataset.

| | |
|---|---|
| Synthetic dialog | Human: What is the proper technique for playing the guitar. Assistant: The proper technique for playing the guitar involves mastering proper hand and body posture, learning to hold the pick correctly, learning basic chords, scales, and music theory, and getting in the habit of regularly practicing. Human:I'm feeling a bit overwhelmed–could you break down the steps more? Assistant: |
| text-davinci-003 response | Absolutely. The first step is mastering proper hand and body posture for holding the guitar. It's important to maintain a good posture in order to prevent fatigue and make sure your hand muscles remain nimble and rested. Then, you'll need to work on specific lead exercises, like frailing or 'down-up' picking. This will help your fingers stay strong, fast and dexterous for chords and solos. Additionally, it's essential to learn about basic chords, chords progression, scales and music theory as that knowledge underpins almost every piece of guitar music. Lastly, consistently practicing will build your technique and, of course, strengthen your skills, stay sharp, and improve your tone. |
| text-davinici-002 response | 1. Master proper hand and body posture. 2. Learn to hold the pick correctly. 3. Learn basic chords, scales, and music theory. 4. Practice regularly. |
| GPT-NeoX-20B response | Absolutely. First, it's helpful to get comfortable with a pick and know where it should be stored. Next, you should make sure you use the right posture for playing the guitar. This can be achieved through stretching before playing, or using an instructional program after playing. After that's done, you will need to learn some basics of chords. And last but not the least, find ways to regularly get in practice. Does have an easier time understanding what you mean? |
| Pythia 6.9B response | Certainly, I can! Step 1: Familiarize yourself with the parts of the guitar. Step 2: Set a realistic goal for the project and step 3: Find the right materials. Step 4: Start building your first album. Step 5: Don't worry if you don't have all the instruments and need the help to get started. |
| Pythia 1.4B response | Absolutely! First, it's important to have a good eye-in-air position during which the end of the pick stands at you, rather than leaning at either shoulder and your back. Then begin by learning a few chord progressions, such as major, minor, and major and minor. In the process, use shifts and fingerings and practice using scales and arpeggios to create a better melodic or harmonic feel. Finally, use practice and repetition to solidify your skills in general chords. |
| Pythia 125M response | Practicing is key, when you can make more on your practice day as well as taking the proper stance. Is there anything I can help you with? |

Table 7: An example of a synthetically generated multi-turn dialog and responses from different model sizes.