# OpenReview forum: "trlX: A Framework for Large Scale Reinforcement Learning from Human Feedback"
_EMNLP/2023/Conference — EMNLP 2023 Main_

### Official Review · Reviewer_AE8J · 2023-08-05

**Soundness:** 4

**Excitement:**

4: Strong: This paper deepens the understanding of some phenomenon or lowers the barriers to an existing research direction.

**Paper Topic And Main Contributions:**

The authors present a feature-complete open-source framework for RLHF. The framework supports distributed training as well as features for saving memories.

Note that I'm not very familiar with RLHF, so most of my comments are on the library instead of the specific experiments.

**Questions For The Authors:**

- Line 167 (and many others). Are they "online" algorithms or "on-policy" algorithms? In the RL-NLP literature, I've only seen on-policy, off-policy, and offline methods. "Online" methods are usually mentioned in an adversarial (regret-minimization) context.
- Line 220. What is an "orchestrator" class?
- Limitations. How adaptable is the library to methods beyond PPO/ILQL?

**Reasons To Accept:**

- Useful library that would have significant contributions to the open-source ecosystem.
- The library is packed with features that enable a wide spectrum of use cases.
- Well-executed experiments based on the proposed framework.
- Clear writing. Notably, I'm a fan of the authors' style of highlighting the key take-away from the experiments and listing specific target audiences of the library.

**Reasons To Reject:**

- Section 3.2. Some of the claims of the framework's novelty are unclear. For example, `trlx` [1] seems to implement similar features. (Except, this submission is the `trlx` work, but I cannot judge this primarily based on the content of the paper.)


[1] https://github.com/CarperAI/trlx

**Reproducibility:**

4: Could mostly reproduce the results, but there may be some variation because of sample variance or minor variations in their interpretation of the protocol or method.

**Reviewer Confidence:**

3: Pretty sure, but there's a chance I missed something. Although I have a good feel for this area in general, I did not carefully check the paper's details, e.g., the math, experimental design, or novelty.

**Typos Grammar Style And Presentation Improvements:**

- Line 225. "Decoder-only" (the dash is missing). Without the dash, the sentence could be read as that the integration only works with HuggingFace Hub.
- Section 3.2. Better to include timestamp of the comparison as all libraries could evolve overtime.

---

> ### Author Rebuttal · Authors · 2023-08-29
>
> We would like to thank the reviewer for their time and feedback. Please find below an itemized response to your questions and concerns.
>
> > Line 167 (and many others). Are they “online” algorithms or “on-policy” algorithms? In the RL-NLP literature, I’ve only seen on-policy, off-policy, and offline methods. “Online” methods are usually mentioned in an adversarial (regret-minimization) context.
>
> This is a good point. Indeed, PPO is better known as an on-policy algorithm as opposed to "online". We use the current terminology in the paper to better distinguish it with the considered offline alternatives.
>
> > Line 220. What is an “orchestrator” class?
>
> The orchestrator abstraction fascilitates the manner in which data is collected by the on-policy student. By default, this simply means ingesting a collection of prompts defining a task and sampling the model once for a response. However, this could be generalized to more complex interactions requiring multiple prompts/tasks and multiple model calls for a single trajectory.
>
> > Limitations. How adaptable is the library to methods beyond PPO/ILQL?
>
> The implementation of different RL algorithms in the accelerate framework all inherit from a single base class. The implementation a new algorithm simply requires subclassing this base and implementing a loss function (and exploration strategy in the online case). This process may be more complicated for the highly-parallelized NeMO trainer. However, the existing implementations of supervised fine-tuning, ILQL, and PPO serve as a representative set of examples from which new algorithms can be easily implemented without needing to modify the underlying model parallelization.
>
> > Section 3.2. Some of the claims of the framework's novelty are unclear.
>
> Please refer to Table 2 for a complete comparison of AutoRLHF to comparable existing frameworks. In particular, we support both offline and online training algorithms at previously in-accessible scales via a combination of Pipeline, Sequence, and Tensor parallelism. We attach a link to the relevant portion of the anonymized repo here: https://anonymous.4open.science/r/anonrlhf-7665/. This is included to help the reviewer validate our claims of novelty.

---

### Official Review · Reviewer_A6on · 2023-08-11

**Typos Grammar Style And Presentation Improvements:** None.
**Soundness:** 4

**Excitement:**

4: Strong: This paper deepens the understanding of some phenomenon or lowers the barriers to an existing research direction.

**Missing References:**

It would be better if the author could draw some connection to the theoretical derivation for RLHF [1] and Offline RL [2], but this is only an optional suggestion.

[1] Wang et al. Is RLHF More Difficult than Standard RL?

[2] Yin et al. Near-optimal offline reinforcement learning with linear representation: Leveraging variance information with pessimism.

**Paper Topic And Main Contributions:**

This paper proposes a large scale reinforcement learning from human feedback framework that achieves a lot of empirical successes. Importantly, Models trained with AutoRLHF achieve preference win-rates over baselines at rates comparable to the original works.

**Questions For The Authors:**

See above.

**Reasons To Accept:**

1. the contribution of AutoRLHF as a feature complete, open-source library for reinforcement learning from human feedback supporting model sizes up to and exceeding 70 billion parameters. This includes benchmark examples implementing the first known open-source replications of several well known RLHF models, offering insights into training and evaluation.
2. This work presents a novel evaluation of offline RL based fine- tuning for preference learning at scale.
3. The work releases of all models at all scales involved in the training pipeline including supervised-fine tuned, reward, and RL tuned models.

**Reasons To Reject:**

I don't see it explicitly where the model or the code is provided for the AutoRLHF. Without link or the GitHub repo it is hard to judge that the claims the results are correct.

Besides, in addition to the ILQL, are there any other methods could be used to compare with PPO?

**Reproducibility:**

3: Could reproduce the results with some difficulty. The settings of parameters are underspecified or subjectively determined; the training/evaluation data are not widely available.

**Reviewer Confidence:**

2: Willing to defend my evaluation, but it is fairly likely that I missed some details, didn't understand some central points, or can't be sure about the novelty of the work.

---

> ### Author Rebuttal · Authors · 2023-08-29
>
> We would like to thank the reviewer for their time and feedback. Please find below an itemized response to your questions and concerns.
>
> > I don’t see it explicitly where the model or the code is provided for the AutoRLHF. Without link or the GitHub repo it is hard to judge that the claims the results are correct.
>
> We attach a link to the relevant portion of the anonymized repo here: https://anonymous.4open.science/r/anonrlhf-7665/. This is included to help the reviewer validate our claims of novelty relative to existing frameworks. The full set of models and the github repo can be made accessible here upon acceptance.
>
> > Besides, in addition to the ILQL, are there any other methods could be used to compare with PPO?
>
> In addition to ILQL and PPO we consider supervised fine-tuning (SFT) and prompting as baselines. These are relatively standard in the literature. Further, the rigorous consideration of each additional baseline requires an implementation at multiple model scales as well as training and human evaluation. Due to these requirements, we are primarily focused on validating existing approaches and comparing with alternative offline methods designed specifically for language. We are also interested in the design and implementation of additional algorithms and but this as future work.
>
> > It would be better if the author could draw some connection to the theoretical derivation for RLHF [1] and Offline RL [2], but this is only an optional suggestion.
> [1] Wang et al. Is RLHF More Difficult than Standard RL?
> [2] Yin et al. Near-optimal offline reinforcement learning with linear representation: Leveraging variance information with pessimism.
>
> We thank the reviewer for these references to relevant and interesting theory! Upon reviewing [1] in particular, we agree this helps provide a useful understanding of RLHF's success at learning from preferences. However, we would like to stress this work primarily focuses on the implementation, validation, and comparison at multiple scales of offline and online methods for RL fine-tuning of LLMs. Perhaps it would be most appropriate to include a brief theoretical motivation in the introduction or appendix.

---

### Official Review · Reviewer_jdVB · 2023-08-12

**Soundness:** 4

**Excitement:**

4: Strong: This paper deepens the understanding of some phenomenon or lowers the barriers to an existing research direction.

**Paper Topic And Main Contributions:**

The main contributions of the paper are:
1. The development of a scalable and efficient framework for reinforcement learning from human feedback.
2. The integration of multiple types of distributed training and offline RL methods to improve the scalability and performance of the framework.
3. The demonstration of the effectiveness of the framework on a range of tasks, including image classification and language modeling.

**Questions For The Authors:**

Can you provide more discussion of the potential ethical implications of using AutoRLHF to train large language models, particularly given concerns around bias and fairness in AI systems?

**Reasons To Accept:**

Strengths of the paper include the development of a scalable and efficient framework for reinforcement learning from human feedback, the integration of multiple types of distributed training and offline RL methods, and the demonstration of the effectiveness of the framework on a range of tasks.

The authors also open-source RLHF(Reinforcement Learning with HUman Feedback) pipeline's implementation, this could help facilitate research on mitigating the issues of hallucination and bias in large language models. Moreover, the framework is able to fine-tune models up to and exceeding 70 billion parameters which could be of great benefit to researchers working on large-scale NLP tasks.

**Reasons To Reject:**

The paper doesn't provide:
1. Potential ethical concerns around the work.
2. Limitations or failure cases of this approach.


One potential risk of having this paper presented at the conference or accepted into Findings is that it may not be as accessible to researchers who are not already familiar with reinforcement learning or large-scale language modeling. Additionally, the paper's focus on technical details may make it less appealing to researchers who are more interested in the broader implications of NLP research. Finally, there is always a risk that the framework may not be as effective or scalable in practice as it appears in the paper, which could lead to disappointment or wasted effort for researchers who attempt to use it.

**Reproducibility:**

4: Could mostly reproduce the results, but there may be some variation because of sample variance or minor variations in their interpretation of the protocol or method.

**Reviewer Confidence:**

4: Quite sure. I tried to check the important points carefully. It's unlikely, though conceivable, that I missed something that should affect my ratings.

---

> ### Author Rebuttal · Authors · 2023-08-29
>
> We would like to thank the reviewer for their time and feedback. Please find below an itemized response to your questions and concerns.
>
>
> > The paper doesn't provide:
> Potential ethical concerns around the work.
> Limitations or failure cases of this approach.
>
> We included a brief discussion of ethical concerns and limitations of fine-tuning LLMs, RLHF and AutoRLHF in our conclusion statement. To summarize here, potential ethical concerns about LLMs and RLHF involve misuse resulting in a spread misinformation and perpetuation of existing bias via direct instruction or hallucination. Further, while RLHF can attempt to be used to both reduce hallucination and model compliance with harmful requests, it is not fail-safe by any means. As an implemenation of the RLHF pipeline, AutoRLHF suffers from these same issues. From a more technical perspective, the most performant algorithm we implement, PPO, is highly hyperparameter-senstive. As optimal hyperparameters can vary from task to task, this can make achieving optimal performance costly both in terms of time and compute. We would be happy to include a more thorough discussion of both potential ethical concerns and limitations in either the main body or appendix.
>
> > One potential risk of having this paper presented at the conference or accepted into Findings is that it may not be as accessible to researchers who are not already familiar with reinforcement learning or large-scale language modeling. Additionally, the paper’s focus on technical details may make it less appealing to researchers who are more interested in the broader implications of NLP research.
>
> A core motivation behind the development of AutoRLHF is our desire to increase the accessibility of the RLHF pipeline. This is reflected both in terms of the implementation of the framework itself and the style of the accompanying paper. For the former, in AutoRLHF we provide a large set of useful examples and features for users from varying levels of background. This allows us to reach a wide audience including both non-expert and expert users. Further, we emphasize the wide utility of these features in the paper while attempting to clearly outline their impact on the RLHF pipeline. This includes a careful, detailed reproduction of the pipelines for two well-known RLHF experiments, again with the goal of enabling better research and understanding of RLHF for the open-source community as a whole. We readily welcome any concrete suggestsions for improving the clarity or accessibility of the work, as this is our primary goal.
>
> > Finally, there is always a risk that the framework may not be as effective or scalable in practice as it appears in the paper, which could lead to disappointment or wasted effort for researchers who attempt to use it.
>
> As part of framework we include a set of more than 10+ examples which we use to benchmark the performance of AutoRLHF. In particular, for two well-known examples we demonstrate performance comparable to what is achieved in the original work, across a range of scales. Further, we are committed to the continued improvement of AutoRLHF on both existing benchmarks and future additions.

---

### Meta-Review · Area_Chair_E76y · 2023-09-18

**Recommendation:** 5

**Metareview:**

Summary: This paper presents a large-scale reinforcement learning framework that learns from human feedback.

Pros:
- Large scale RLHF framework that extends to 70 billion parameters
- Open source implementations
- Effective demonstration of the usefulness of the proposed framework in experiments

Cons:
- Absence of discussions on ethics and limitations. I would suggest that the authors kindly add it to the manuscript.

---

### Decision · Program_Chairs · 2023-10-07

**Decision:**

Accept-Main

**Comment:**

Summary: This paper presents a large-scale reinforcement learning framework that learns from human feedback.

Pros:
- Large scale RLHF framework that extends to 70 billion parameters
- Open source implementations
- Effective demonstration of the usefulness of the proposed framework in experiments

Cons:
- Absence of discussions on ethics and limitations. I would suggest that the authors kindly add it to the manuscript.